# Associations between Physical Activity and Food Intake among Children and Adolescents: Results of KiGGS Wave 2

**DOI:** 10.3390/nu11051060

**Published:** 2019-05-11

**Authors:** Kristin Manz, Gert B. M. Mensink, Jonas D. Finger, Marjolein Haftenberger, Anna-Kristin Brettschneider, Clarissa Lage Barbosa, Susanne Krug, Anja Schienkiewitz

**Affiliations:** Department of Epidemiology and Health Monitoring, Robert Koch Institute, 13353 Berlin, Germany; MensinkG@rki.de (G.B.M.M.); FingerJ@rki.de (J.D.F.); HaftenbergerM@rki.de (M.H.); BrettschneiderA@rki.de (A.-K.B.); Lage-BarbosaC@rki.de (C.L.B.); KrugS@rki.de (S.K.); SchienkiewitzA@rki.de (A.S.)

**Keywords:** Physical activity, exercise, food intake, diet, children, adolescents, KiGGS

## Abstract

A balanced diet and sufficient physical activity are essential for the healthy growth of children and adolescents and for obesity prevention. Data from the second wave of the population-based German Health Interview and Examination Survey for Children and Adolescents (KiGGS Wave 2; 2014–2017) were used to analyse the association between food intake and physical activity among 6- to 17-year-old children and adolescents (*n* = 9842). Physical exercise (PE) and recommended daily physical activity (RDPA) were assessed with self-administered questionnaires and food intake by a semi-quantitative food frequency questionnaire. Multivariable logistic regression was used to analyse the association between food group intake (dependent variable) and level of PE or RDPA. High levels of physical activity (PE or RDPA) were associated with higher consumption of juice, water, milk, dairy products, fruits, and vegetables among both boys and girls, and among boys with a higher intake of bread, potatoes/pasta/rice, meat, and cereals. Higher PE levels were also less likely to be associated with a high soft drink intake. High levels of RDPA were associated with high intake of energy-dense foods among boys, which was not observed for PE. This study indicates that school-aged children and adolescents with higher levels of physical activity consume more beneficial foods and beverages compared to those with lower physical activity levels.

## 1. Introduction

A balanced diet and sufficient physical activity are essential for a healthy growth and development of children and adolescents and are important determinants of health throughout the life course. Unbalanced dietary patterns (with high amounts of highly processed and energy dense foods) are associated with unfavourable cardiometabolic risk factors (e.g., blood pressure, blood glucose, insulin levels, and lipid profile) among adolescents [1,2]. Higher levels of physical activity are associated with better physical, psychological, and cognitive health of children and adolescents [3], whereas a predominantly sedentary lifestyle is associated with less favourable levels of cardiometabolic risk factors [4]. In addition to genetic predisposition [5], dietary behaviour and physical activity are important determinants of obesity [6,7].

The prevalence of childhood obesity has significantly increased in recent years and is, therefore, a priority for health promotion and prevention policy and action [8]. Current data from KiGGS wave 2 (2014–2017) show that 15% of the 3- to 17-year-old children and adolescents in Germany are overweight; almost 6% are obese [9]. Compared to the KiGGS baseline survey (2003–2006), the prevalence of overweight and obesity has not increased further in the last ten years among children and adolescents, but remains at a high level [9].

There are only a few studies that have comprehensively analysed the relation between the two key components of obesity development, dietary behaviour, and physical activity in children and adolescents. In two studies, one conducted among high school students in the USA and the other among 12-year-old children from France, a higher level of physical activity is associated with a higher intake of fruits and vegetables [10,11]. The international study ISCOLE among 9- to 12-year-old children from 12 countries showed that achieving a higher number of specific physical activity recommendations (for sedentary behaviour, physical activity, and sleep duration) is associated with a more preferable dietary pattern [12]. A review of several studies investigating obesogenic behaviours using cluster analyses indicated that diet, physical activity, and sedentary behaviour cluster in both unhealthy and healthy ways and that cluster patterns differ by sex, age, and socioeconomic status [6].

Most existing studies investigating the relation between physical activity and dietary behaviour cover just a small age-range of childhood and do not indicate how the intake of single food items differ by the level of physical activity. Indicators of the current physical activity levels, as well as consumption of particular food items of children and adolescents have been described for Germany [13,14]. However, the interrelationship between these two behavioural aspects has not yet been analysed.

The aim of this study is to analyse whether children and adolescents aged 6 to 17 years with higher levels of physical activity differ in their food intake compared to those with lower levels of physical activity. 

## 2. Materials and Methods

### 2.1. Study Design and Study Population

KiGGS (German Health Interview and Examination Survey for Children and Adolescents) is part of the Federal Health Monitoring System of the Robert Koch Institute and consists of regularly conducted representative cross-sectional surveys among children and adolescents aged between 0 and 17 years living in Germany. KiGGS Wave 2 was conducted between 2014 and 2017. The design and methodology of KiGGS Wave 2 have been described in detail elsewhere [15,16]. In brief, the study sample was drawn from the German resident population aged 0 to 17 years using a two-stage cluster sampling approach. Firstly, 167 sample points were randomly selected proportional to the population densities in the federal states and community sizes within the Federal Republic of Germany. Secondly, within each sample point age-stratified samples of individuals were randomly selected from the local population registries. In total, 15,023 children and adolescents (7538 girls; 7,485 boys) participated in the cross-sectional survey of KiGGS Wave 2.

KiGGS Wave 2 was conducted in accordance with the Declaration of Helsinki, and the protocol was approved by the Federal Commissioner for Data Protection and Freedom of Information and by the ethics committee of the Hannover Medical School (Number 2275–2014). A written informed consent was obtained by parents and all participants aged 14 years and above before data collection.

### 2.2. Data Collection and Aggregation

In KiGGS, parents (or legal guardians) of the 3- to 10-year-olds completed self-administered questionnaires which provided information on their children’s health and health behaviour (e.g., food intake and physical activity). Similar questionnaires were filled in by 11- to 17-year-olds themselves.

### 2.3. Food Intake

KiGGS Wave 2 included a self-administered semi-quantitative food frequency questionnaire (FFQ) to assess the consumption of selected food items [17]. The FFQ starts with a short written introduction about the questionnaire. Participants are instructed how to report the frequency and number of standard portions which were consumed during the last four weeks. An illustration is provided as an example to report two slices of bread that were eaten twice a day during the past four weeks. A further illustration shows how to make corrections in case this is necessary. Finally, a telephone number was provided for further questions about the completion of the questionnaire. The 11- to 17-year-old participants or parents of 3- to 10-year-old participants were asked about the food items they or their children had consumed ‘during the last four weeks’. The questionnaire contained 48 food items for younger participants and 53 food items (including alcoholic beverages) for those aged 11 years and above. First, the consumption frequency of each food item was assessed with the question: ‘How often did your child/did you eat/drink food/drink X?’ and in most cases, examples of the particular food items were given. Answer categories were: ‘never’, ‘once per month’, ‘2–3 times per month’, ‘1–2 times per week’, ‘3–4 times per week’, ‘5–6 times per week’, ‘daily’, ‘2 times per day’, ‘3 times per day’, ‘4–5 times per day’, ‘more than five times per day’. Second, portion size information was obtained by questions following the pattern: ‘If your child/you eat/drink food/drink X, how much does your child/do you usually eat/drink?’ To support portion size estimation, pictures were included as a reference for predefined portion sizes. In each case, five answering categories were given, which varied depending on the food item. These were, for instance, ‘½ a glass (or less)’, ‘1 glass’, ‘2 glasses’, ‘3 glasses’, and ‘4 glasses (or more)’. For some food items, a specific aspect of the food consumption was inquired, such as the dilution ratio of juice and water.

For the presented analyses, the information on food consumption frequencies was transformed into the number of occasions of consumption of each food over a four-week period (28 days). The portions were converted into grams or millilitres. For each food item, the estimated amount consumed per day was obtained by multiplying the converted food consumption frequency and the portion amount, and dividing this by 28. The following food groups were used for the analyses, partly constructed by summarising the amounts of multiple food items (in parenthesis): Soft drinks (sugar-sweetened soft drinks; reduced calorie soft drinks), juices (fruit juices; vegetable juices), water, milk, dairy products (cream cheese; cheese; quark, yoghurt or soured milk), fast food (hamburger or doner kebab; grilled sausages; French fries; pizza), fruits (fresh fruits; processed fruits), vegetables (raw vegetables; legumes; cooked vegetables), bread (whole-grain bread or rolls; brown or mixed bread or rolls; white bread or rolls), potatoes, noodles & rice (cooked potatoes; fried potatoes; noodles; rice), meat (poultry; red meat; meat portion in hamburger or doner kebab; grilled sausages; sausage; ham), breakfast cereals (corn flakes; muesli), savoury snacks (potato crisps; salty snacks or crackers), confectionery (cakes, tartes or sweet pastries; biscuits; chocolate; sweets; ice cream; sweet spreads). Based on the consumed amounts of these food groups, participants were ranked into quintiles. Quintiles were calculated separately for boys and girls in the age groups 6–10, 11–13 and 14–17 years in order to account for gender- and age-related differences in energy and nutrient requirements. For each food group, the top two quintiles (40%) were assigned to the category ‘high intake’. A table with the threshold values for the category ‘high intake’ can be found in the Appendix A.

### 2.4. Recommended Daily Physical Activity (RDPA) and Physical Exercise (PE)

In KiGGS Wave 2, RDPA and PE information was also obtained using self-administered questionnaires. The indicator RDPA describes the amount of days per week with physical activity (light to vigorous intensity) of at least 60 min. To assess RDPA, participants were asked ‘How many days of a normal week are you/is your child physically active for at least 60 min on a single day?’ The eight answer categories ranged from ‘Seven days’ to ‘None’. The answers were assigned to the categories ‘low’ (one hour/day on less than 3 days/week), ‘medium’ (one hour/day on 3 to 5 days/week) and ‘high’ (one 1 hour/day on 6 to 7 days/week). The instrument was initially used to verify if participants met the ‘at least 60 min daily physical activity’ criteria recommended by the World Health Organization [18] and was adapted from Prochaska and colleagues [19] A validation study showed moderate test-retest reliability with a Kappa coefficient of 0.54 and a significant Spearman correlation coefficient of 0.24 for validity when compared with objectively measured physical activity by accelerometers [20]. The indicator PE should include sports activities like running or playing soccer. To obtain information on PE, participants were asked whether they participate in physical exercises, adding that this referred to all kinds of sports regardless if it was within a sports club or not. Physical education at school should not be included. The question could be answered with ‘Yes’ or ‘No’. Participants who answered ‘Yes’ were consecutively asked how many minutes or hours they usually participate in physical exercises per week. The data on PE was used to assign the persons to categories ‘low’ (less than one hour of sport/week), ‘medium’ (1–3 h/week) and ‘high’ (more than 3 h/week).

### 2.5. Socioeconomic Status (SES)

In the KiGGS study, an index is used to measure parental socioeconomic status (SES), which is based on information about the parents’ education, their occupational status and income [21,22]. A detailed description of the construction of the socioeconomic variables and the SES index can be found elsewhere [22]. Based on the SES index, individuals were categorized into ‘low’, ‘medium’ and ‘high’ SES.

### 2.6. Body Mass Index (BMI)

In the physical examination component, standardized measurements of body height and weight were obtained [9]. The body mass index (BMI) was calculated from body weight divided by the square of body height. BMI percentiles were modelled as a function of age (BMI-for-age) and transformed to a standard normal distribution (BMI-for-age *z*-score). 

### 2.7. Data Analysis

Since preschool children have a considerably different physical activity and food consumption pattern from older children, 3- to 5-year-olds were not included in the present analysis. The current analysis was, therefore, restricted to 11,014 participants aged 6 to 17 years (5582 girls and 5432 boys). 

Food consumption information was completely set to missing for some cases in the original dataset (*n* = 38). This was done if the frequency answers contained more than 20 missing values or if the calculated total intake was implausibly high (total daily amount of beverages exceeded 15 litres, of solid foods exceeded 10 kg or if there was a combination of the amount of beverages exceeding 4 L and solid foods exceeding 6 kg). In the current analysis, participants with missing information on at least one of the aggregated food intake variables were excluded from the current analysis (*n* = 1172, 10.6%). The sample for multivariable regression analyses among boys comprises 4472 participants for PE and 4557 participants for RDPA, after excluding participants with missing information on PE, RDPA or SES. Among girls, the sample for the multivariable regression comprises 5016 for PE and 5073 participants for RDPA.

Statistical analyses were conducted using Stata SE 15 (StataCorp. 2017. Stata Statistical Software: Release 15. StataCorp LLC, College Station, TX, USA). All analyses were performed with survey design procedures to adjust for the clustered sampling design. A weighting factor was applied to correct deviations within the sample from the German population with regard to age, gender, federal state (as of 31.12.2015), nationality (as of 21.12.2014), and the parents’ level of education (Mikrozensus 2013). Multivariable logistic regression was used to analyse if medium and high versus low RDPA and PE were associated with food intake variables. Odds ratios (OR) were reported to describe the odds of having a high intake of a food depending on the variables PE and RDPA. The associations were stepwise adjusted for age (model 1) and parental SES (model 2). In a sensitivity analysis, BMI-for-age *z*-scores were added to model 2. The criterion for statistical significance was set at *p* < 0.05.

## 3. Results

Demographic characteristics and physical activity behaviour of the study sample stratified by sex, age, and SES are shown in Table 1. PE was low among 25.7%, medium among 36.7% and high among 37.6% of the participants, while RDPA was low among 25.2%, medium among 47.8% and high among 27.0% of the participants. Boys more often had a high level of PE and RDPA than girls. The proportion of boys and girls with a high PE level was higher among older age groups, whereas the proportion with a high RDPA level was lower among older age groups. The proportion of boys and girls with a high PE level increased with increasing SES, while there were no statistically significant differences in the frequency of high RDPA levels by SES. However, the proportion of participants with low RDPA significantly decreased with increasing SES.

### 3.1. Binary Associations

A higher PE level was significantly associated with a high intake of water, dairy products, fruits, and vegetables in boys and girls, and a high intake of juice and breakfast cereals in boys (Table 2). A medium or high PE level was less often associated with a high soft drink intake than a low PE level among boys and girls. Furthermore, a low PE level was more often associated with a higher meat intake for girls compared to medium or high PE level, while a low PE level was associated with a high intake of savoury snacks for both genders. 

Higher RDPA was significantly associated with a high intake of fruits and vegetables among boys and girls. Furthermore, among boys, higher levels of RDPA were associated with a high intake of water, milk, dairy products, bread, potatoes/pasta/noodles, breakfast cereals, and confectionery (Table 2). Girls with a low RDPA level more often had a high intake of soft drinks, savoury snacks, and confectionery than girls with a medium or high RDPA level. 

### 3.2. Multivariable Analyses

Almost all significant associations in the binary analyses for PE remained significant in the logistic regression analyses after adjustment for age and SES (model 2; Table 3). Exceptions were the associations of PE with water and with meat among girls. Furthermore, the association of PE and a high intake of breakfast cereals and savoury snacks among girls became significant after adjusting for age, but did not remain significant after further adjustment for SES.

For boys, a high compared to a low level of PE was less likely associated with a high soft drink intake (OR 0.7), but was more likely associated with a high intake of juice (OR 1.3), water (OR 1.3), dairy products (OR 1.5), fruits (OR 1.9), vegetables (OR 1.7), and breakfast cereals (OR 1.5) (Table 3). For girls, a high compared to a low level of PE was less likely associated with a high intake of soft drinks (OR 0.7), and more likely associated a high intake of dairy products (OR 1.2), fruits (OR 1.6), and vegetables (OR 1.5).

Many binary associations of RDPA with a certain food group intake remained significant in the multivariable analyses (Table 4). Among boys, the non-significant association observed for a high intake of juice and meat with higher levels of RDPA became significant after adjustment for age and SES. For girls, the significant binary association between RDPA and high soft drink intake did not remain significant after adjusting for age and SES, whereas the associations between RDPA and high intake of juice, water, milk, and dairy products became significant.

For boys, a high compared to a low RDPA was more likely associated with a high intake of juice (OR 1.3), water (OR 1.4), milk (OR 1.7), dairy products (OR 1.4), fruits (OR 2.1), vegetables (OR 1.5) bread (OR 1.5), potatoes/pasta/rice (OR 1.4), meat (OR 1.3), breakfast cereals (OR 1.3), and confectionery (OR 1.6) for boys (Table 4). For girls, a high compared to a low RDPA was more likely associated with a high intake of water (OR 1.3), milk (OR 1.3), dairy products (OR 1.4), fruits (OR 1.9), and vegetables (OR 1.5). A medium compared to a low RDPA was less likely associated with a high intake of savoury snacks (OR 0.8) and confectionery (OR 0.8), and more likely associated with a high intake of juice (OR 1.3) for girls.

The inclusion of BMI-for-age *z*-scores into model 2 did not alter the relation between PE nor RDPA and food intake (data not shown).

## 4. Discussion

Adequate levels of physical activity and a balanced diet are important requirements for an optimal physiological and cognitive development, as well as for the maintenance of good physical conditions and health in general. A good balance between physical activity and diet may also help to prevent or reduce the occurrence of obesity [23]. For a holistic prevention approach, it is important to understand and take into account that these health behaviours may interact with each other. The aim of the current study was, therefore, to analyse differences in intake of various food groups between low, medium and high physical activity behaviour groups among children and adolescents. To our knowledge, this study is the first study to investigate the associations between two different aspects of physical activity behaviour and 14 different food groups based on representative data from children and adolescents living in Germany. These analyses among boys and girls aged 6 to 17 years indicate that higher physical activity levels are more often associated with a high intake of predominantly preferable foods (with some exceptions) and a less frequent high intake of some less beneficial foods. For example, boys and girls with a high RDPA level were twice as likely to consume fruits in high amounts and had 50% higher odds to consume vegetables in high amounts than those with a lower RDPA. Furthermore, the odds ratio of a high soft drink intake was 30% lower in children and adolescents with a high compared to a low PE level. 

The more often observed high intake of several food groups among children and adolescents with higher physical activity levels (PE and RDPA), some of which are dairy products, fruits, and vegetables, may partly be explained by a higher energy requirement due to a higher physical activity. A derivation of accurate estimates of the energy intake is not possible due to the relatively short food frequency questionnaire used in KiGGS Wave 2. Nevertheless, a high RDPA level was more often associated with a high intake for many food groups for boys, which may suggest a higher energy intake for this group. Among girls, high intakes of some food groups, i.e., for water, fruits, and vegetables, were associated with higher RDPA levels, although these are not energy-dense foods. Overall, this suggests that differences in energy needs may not explain all observed differences in food intake. The observed association of high intake of water and higher physical activity levels may be related to higher transpiration losses during enhanced physical activity. The authors of a review summarised that the relation between energy expenditure caused by physical activity and energy intake in children and adolescents are still inconclusive due to a lack of data [24]. However, it was observed that nutritional adaptation occurs as a response to physical activity, not only to compensate for the expended energy [24,25]. These adaptations may be changes in food choices as well as in appetite sensation, depending on duration, intensity, and type of physical activity [26,27]. Differences in duration, intensity and type of physical activity between boys and girls might also explain why, in this present analysis, more food groups were associated with physical activity in boys compared to girls. The higher levels of RDPA and PE among boys than girls [13] might have a stronger impact on their food choices. Furthermore, our results confirm that different types of physical activity are differently associated with a high consumption of food items: RDPA was more often associated with a high intake of individual foods compared to PE. Various reasons may lead to engagement in more PE or RDPA. These include body weight loss or maintenance, to gain muscle power, to improve body fitness, or physical skills as well as social aspects [28]. These purposes may differ between genders as well as for RDPA and PE, and could partly explain the observed differences in associations with food intake. 

Studies comparing the intake of different food groups and physical activity in children and adolescents are rare. The observed higher fruit and vegetable consumption in children with higher physical activity is in line with studies among school-aged children in the USA and France [10,11]. In a representative sample of adolescents attending grade 9 to 12 in the USA, a higher fruit, vegetable, and soft drink consumption was associated with higher daily physical activity [10]. Regarding the soft drink intake, our study showed contradictory results: a lower soft drink intake was associated with a high PE level. There was no significant association observed between a high soft drink intake and RDPA. A study among 12-year-olds from France showed that the consumption of fruits, vegetables, and fruit juice was positively associated with organised physical activity [11]. In our study, in addition to the observed association with fruit and vegetable intake, fruit juice was also positively associated with PE and RDPA in boys and with RDPA in girls.

Most studies regarding physical activity and food intake in children summarise the assessed food intake into the categories ‘healthy dietary pattern’ and ‘unhealthy dietary pattern’. If we apply these categories to the food groups presented in our study, a high intake of water, fruits, and vegetables would be considered as components of a healthy dietary behaviour, whereas a high intake of soft drinks, fast food, savoury snacks, and confectionery would be unhealthy [29]. From this perspective, the current results suggest that a high PE level is more likely associated with a healthy food pattern, reflected by high intakes of fruit and vegetables and low intakes of soft drinks. High RDPA also seems to be associated with more beneficial intakes of fruits, vegetables, and drinking water. Additionally, high RDPA seems to be associated with higher intakes of energy-dense food groups and less beneficial food groups, like confectionery and meat, among boys. Among girls, higher RDPA was associated with a lower intake of savoury snacks and confectionery. The positive association between PE and a healthy food intake might be explained by a higher overall health consciousness which influences multiple health behaviours [30,31]. For younger children, the motivation of the parents and family to live a healthy lifestyle will be relevant rather than the individual preference [32].

The authors of a review about the clustering of physical activity, sedentary behaviour and diet in children and adolescents concluded that these behaviours show inconsistent clustering patterns, sometimes corresponding with healthy behaviours and sometimes not [6]. Two studies found a consistency in preferable physical activity and dietary behaviour which confirms our results. However, in a cluster analysis conducted with children and adolescents aged 11 to 17 years living in Germany no cluster with high scores in physical activity and healthy diet was observed [33].

Data of the International Study of Childhood Obesity, Lifestyle and the Environment (ISCOLE) were used to examine if meeting recommendations related to physical activity, screen-time and sleep (≥60 min/day moderate-to-vigorous physical activity; ≤2 h/day screen time; 9 to 11 h/night sleep duration) were associated with dietary patterns among 9- to 11-year-old children from 12 different countries [12]. Results showed that a healthier diet was observed when more recommendations were met, which confirm partly a positive relation between sufficient physical activity and a healthy diet. 

A strength of the study is that it is based on a large nationwide and representative sample of young persons living in Germany. In addition, the information on two different aspects of physical activity, as well as on the intake of many food groups, allows a broad analysis of the association between physical activity and food intake among children and adolescents. Our analyses have some limitations that should be considered. The assessment of physical activity and food intake was based on self-reports which is subject to recall and social desirability bias [34,35]. This may lead to overreporting of healthy behaviours (e.g., physical activity, intake of fruits, and vegetables) and an underreporting of unhealthy behaviours (e.g., intake of soft drinks, snacks, and sweets). The use of objective methods, such as activity monitors, may help to reduce such bias for physical activity. However, bias might still be present, since the measurement of specific physical activities by activity monitors is only possible in combination with a self-reported diary, and the consciousness of activity being measured may also alter actual behaviour [36]. In a free-living population setting, it is hardly possible to objectively measure dietary intake. The food frequency instrument used in this survey is not comprehensive and detailed enough to give a complete and precise overview of food consumption and energy intake. In addition, the specific information on food consumption and physical activity behaviour is gathered from parents for the younger children (up to the age of 11 years) and from those above 11 years themselves, which could have biased the results. Although this is inevitable because young children cannot give reliable information on major parts of the requested items, parents may also not have a complete overview of what their children eat (especially out of home) and how physically active they are. So for the different age groups the level of misreporting could be different. However, we constructed age-specific quintiles for the reported food intakes, so persons are ranked according to reported consumption relative to those within their age group. This may at least reduce in some extent the possible bias introduced by the responder. Furthermore, the association between physical activity and food intake may be confounded by other variables, which we have not considered for adjustment. We adjusted for age and SES because these dimensions could bias the association between physical activity and food intake. In a sensitivity analysis, we additionally adjusted for BMI-for-age *z*-scores. This, however, had no substantial impact on the regression results (data not shown). Due to the cross-sectional nature of the data, causal inferences cannot be derived. Finally, around 10% of the participants were excluded because of missing values in the analysed items which might have biased the results. These excluded participants were more often boys, younger children, and participants with lower SES. 

## 5. Conclusions

The analysis of the association between RDPA and PE with food intake among school-aged children and adolescents indicates that higher levels of RDPA and PE are associated with more beneficial intakes of particular food items, such as a high fruit and vegetable intake and a lower soft drink intake. RDPA is additionally more often associated with high intakes of some energy-dense foods among boys, but not among girls. Detailed information on activity patterns and food intake can make a considerable contribution to focused primary prevention strategies which promote a healthy lifestyle of children and adolescents. 

## Figures and Tables

**Table 1 nutrients-11-01060-t001:** Demographic characteristics for categories of physical activity behaviour of the study sample (*N* = 9842).

		Total	Sex		Age (Years)			Socioeconomic Status ^a^		
			Boys	Girls	6–10	11–13	14–17	Low	Medium	High
	*n*	% 95% CI	% 95% CI	% 95% CI	% 95% CI	% 95% CI	% 95% CI	% 95% CI	% 95% CI	% 95% CI
*n*		100.0 9842	47.2 4646	52.8 5196	32.0 3146	35.2 3464	32.8 3232	12.2 1184	62.3 6065	25.5 2480
**Physical exercise** ^b^										
Low	2295	25.7 24.4–27.0	22.1 20.4–23.8	29.2 27.4–31.1	23.5 21.1–26.0	23.1 21.1–25.2	29.9 27.7–32.1	44.2 40.4–48.1	24.0 22.6–25.6	12.9 11.3–14.7
Medium	3639	36.7 35.3–38.1	32.5 30.7–34.3	40.7 38.8–42.8	48.3 45.7–50.8	36.4 34.1–38.8	27.3 25.5–29.2	30.4 27.1–34.0	37.6 36.0–39.4	40.0 38.0–42.2
High	3654	37.6 36.2–39.1	45.4 43.2–47.7	30.1 28.3–31.9	28.3 26.3–30.4	40.5 38.0–43.1	42.9 40.8-45.0	25.3 22.5–28.5	38.3 36.5–40.1	47.0 44.5–49.6
**Recommended daily physical activity** ^c^										
Low	2305	25.2 23.9–26.6	20.3 18.5–22.2	30.1 28.3–31.9	17.1 15.4–18.9	22.8 20.9–24.9	34.2 32.1–36.4	34.1 30.3–38.2	24.1 22.7–25.7	19.7 17.9–21.6
Medium	4786	47.8 46.4–49.2	48.8 46.9–50.6	46.9 44.9–49.0	44.9 42.5–47.3	50.0 47.6–52.5	48.4 46.1–50.6	40.3 36.4–44.2	48.8 47.0–50.5	52.4 50.0–54.8
High	2639	27.0 25.6–28.4	31.0 29.2–32.9	23.0 21.3–24.9	38.0 35.6–40.6	27.1 24.8–29.6	17.4 15.8–19.2	25.6 22.5–29.1	27.1 25.6–28.7	27.9 26.1–29.9

^a^ 113 missing; ^b^ 254 missing; low: < 1 h/week, medium: 1–3 h/week, high: >3 h/week; ^c^ 112 missing; low: <3 days/week 1 h/day, medium: 3–5 days/week 1 h/day, high: 6–7 days/week 1 h/day; CI—confidence interval.

**Table 2 nutrients-11-01060-t002:** Percentage with high intake of specific foods among 6- to 17-year-old children and adolescents in total and stratified by physical activity variables (binary associations).

		Total		Physical Exercise ^a^				Recommended Daily Physical Activity ^b^		
			Low	Medium	High	*P* *	Low	Medium	High	*P* *
High Intake		%	%	%	%		%	%	%	
Soft drinks	boys	39.7	48.2	36.3	37.2	<0.001	42.4	37.4	40.5	0.128
	girls	38.0	47.3	33.2	33.9	<0.001	42.0	35.3	36.0	0.008
	total	38.8	47.7	34.6	35.8	<0.001	42.2	36.3	38.6	0.003
Juice	boys	34.5	31.1	33.8	37.3	0.012	31.5	34.9	36.2	0.210
	girls	35.5	32.4	36.0	36.0	0.223	33.2	37.1	35.2	0.231
	total	35.0	31.9	35.0	36.8	0.014	32.5	36.0	35.8	0.092
Water	boys	39.7	36.8	37.0	42.1	0.016	35.9	39.2	42.8	0.045
	girls	33.5	33.7	31.1	36.5	0.042	32.2	32.9	36.7	0.118
	total	36.5	35.0	33.7	39.8	<0.001	33.7	36.1	40.2	<0.001
Milk	boys	25.6	25.5	24.7	27.1	0.496	21.2	25.7	28.3	0.015
	girls	29.9	30.9	27.8	32.1	0.124	30.8	29.6	29.4	0.776
	total	27.8	28.6	26.4	29.1	0.170	27.0	27.6	28.7	0.557
Dairy products	boys	39.0	33.2	38.6	42.5	0.001	33.8	39.7	41.1	0.013
	girls	39.3	36.8	38.6	43.1	0.021	36.0	40.1	41.9	0.057
	total	39.1	35.3	38.6	42.7	<0.001	35.1	39.9	41.5	<0.001
Fast food	boys	40.0	36.6	39.0	41.0	0.220	39.8	40.1	39.2	0.928
	girls	41.4	43.6	40.4	39.8	0.219	41.3	42.0	39.6	0.614
	total	40.7	40.7	39.8	40.5	0.852	40.7	41.0	39.4	0.628
Fruits	boys	36.0	27.4	33.5	41.9	<0.001	24.3	37.7	41.3	<0.001
	girls	37.1	33.5	35.0	43.9	<0.001	30.6	38.5	41.7	<0.001
	total	36.5	30.9	34.4	42.7	<0.001	28.1	38.1	41.5	<0.001
Vegetables	boys	38.4	31.0	35.6	44.1	<0.001	32.1	38.9	41.9	0.001
	girls	40.1	35.3	38.7	47.6	<0.001	34.7	42.0	43.8	<0.001
	total	39.3	33.5	37.4	45.5	<0.001	33.7	40.5	42.7	<0.001
Bread	boys	37.4	39.3	34.7	37.2	0.168	33.5	36.6	40.9	0.016
	girls	38.8	39.7	38.5	37.5	0.689	36.9	37.9	41.5	0.180
	total	38.1	39.5	36.8	37.3	0.295	35.5	37.2	41.1	0.008
Potatoes/pasta/rice	boys	38.5	36.7	37.2	39.7	0.417	35.6	36.8	42.6	0.013
	girls	39.2	40.5	38.4	38.6	0.596	37.7	40.6	37.5	0.309
	total	38.8	38.9	37.9	39.2	0.663	36.9	38.7	40.4	0.174
Meat	boys	39.8	37.6	38.9	41.1	0.361	37.7	38.6	42.3	0.172
	girls	41.3	45.4	38.9	40.0	0.013	42.2	40.0	42.3	0.438
	total	40.6	42.1	38.9	40.7	0.162	40.4	39.3	42.3	0.159
Breakfast cereals	boys	36.8	31.2	34.0	41.9	<0.001	32.2	38.1	37.7	0.042
	girls	39.3	36.4	39.2	41.6	0.113	37.3	39.3	42.4	0.104
	total	38.0	34.2	37.0	41.8	<0.001	35.3	38.7	39.7	0.020
Savoury snacks	boys	41.0	43.3	39.9	40.1	0.417	40.8	40.9	40.8	0.999
	girls	41.2	44.5	40.0	39.1	0.080	45.0	38.4	41.4	0.011
	total	41.1	44.0	40.0	39.7	0.033	43.3	39.7	41.1	0.098
Confectionery	boys	35.2	31.4	34.2	37.0	0.109	30.7	34.4	39.0	0.009
	girls	40.6	43.9	39.6	40.1	0.143	44.3	37.5	42.8	0.002
	total	37.9	38.6	37.2	38.2	0.719	38.9	35.9	40.6	0.011

^a^*n* = 9488; boys: *n* = 4472, girls: *n* = 5016; low: <1 h/week, medium: 1–3 h/week, high: >3 h/week; ^b^
*n* = 9630, boys: *n* = 4557, girls: *n* = 5073; low: <3 days/week 1 h/day, medium: 3–5 days/week 1 h/day, high: 6–7 days/week 1 h/day; * *P*-values of test for differences between groups (chi-square).

**Table 3 nutrients-11-01060-t003:** Results of logistic regression analyses between food intake and physical exercise ^a^.

High intake of Food Group		Boys (*n* = 4472) Physical Exercise Level		Girls (*n* = 5016) Physical Exercise Level	
		Medium	High	Medium	High
		OR (95% CI)	OR (95% CI)	OR (95% CI)	OR (95% CI)
Soft drinks	Model ^1^	**0.6 (0.5–0.8)**	**0.6 (0.5–0.8)**	**0.6 (0.5–0.7)**	**0.6 (0.5–0.7)**
	Model ^2^	**0.7 (0.6–0.9)**	**0.7 (0.6–0.9)**	**0.7 (0.5–0.8)**	**0.7 (0.6–0.8)**
Juice	Model ^1^	1.2 (1.0–1.4)	**1.3 (1.1–1.6)**	1.2 (1.0–1.6)	1.2 (1.0–1.5)
	Model ^2^	1.2 (0.9–1.4)	**1.3 (1.0–1.5)**	1.2 (1.0–1.6)	1.2 (1.0–1.5)
Water	Model ^1^	1.0 (0.8–1.2)	**1.2 (1.0–1.5)**	0.9 (0.7–1.1)	1.1 (0.9–1.4)
	Model ^2^	1.1 (0.9–1.3)	**1.3 (1.1–1.6)**	0.9 (0.8–1.1)	1.2 (1.0–1.5)
Milk	Model ^1^	1.0 (0.8–1.3)	1.0 (0.8–1.4)	1.0 (0.8–1.2)	1.1 (0.9–1.3)
	Model ^2^	1.0 (0.8–1.3)	1.0 (0.8–1.4)	1.0 (0.8–1.3)	1.1 (0.9–1.4)
Dairy products	Model ^1^	**1.3 (1.0–1.6)**	**1.5 (1.2–1.8)**	1.1 (0.9–1.3)	**1.3 (1.1–1.6)**
	Model ^2^	**1.3 (1.0–1.7)**	**1.5 (1.2–1.9)**	1.0 (0.9–1.3)	**1.2 (1.0–1.5)**
Fast food	Model ^1^	1.1 (0.9–1.4)	1.2 (1.0–1.5)	0.9 (0.7–1.0)	0.9 (0.7–1.1)
	Model ^2^	1.2 (0.9–1.5)	1.2 (1.0–1.5)	1.0 (0.8–1.1)	0.9 (0.8–1.2)
Fruits	Model ^1^	1.3 (1.0–1.7)	**2.0 (1.5–2.5)**	1.1 (0.9–1.4)	**1.6 (1.3–1.9)**
	Model ^2^	1.2 (1.0–1.6)	**1.9 (1.5–2.4)**	1.1 (0.9–1.4)	**1.6 (1.3–1.9)**
Vegetables	Model ^1^	1.2 (1.0–1.6)	**1.7 (1.4–2.2)**	1.2 (1.0–1.4)	**1.7 (1.4–2.0)**
	Model ^2^	1.2 (0.9–1.5)	**1.7 (1.3–2.1)**	1.0 (0.8–1.3)	**1.5 (1.2–1.8)**
Bread	Model ^1^	0.9 (0.7–1.1)	0.9 (0.7–1.1)	1.0 (0.8–1.2)	0.9 (0.7–1.1)
	Model ^2^	0.9 (0.7–1.1)	0.9 (0.8–1.1)	1.0 (0.8–1.2)	0.9 (0.8–1.2)
Potatoes/pasta/rice	Model ^1^	1.1 (0.8–1.4)	1.1 (0.9–1.4)	0.9 (0.8–1.1)	0.9 (0.8–1.1)
	Model ^2^	1.1 (0.8–1.3)	1.1 (0.9–1.4)	1.0 (0.8–1.1)	0.9 (0.8–1.2)
Meat	Model ^1^	1.1 (0.9–1.4)	1.1 (0.9–1.4)	**0.8 (0.7–1.0)**	**0.8 (0.7–1.0)**
	Model ^2^	1.1 (0.9–1.5)	1.2 (1.0–1.5)	0.8 (0.7–1.0)	0.9 (0.7–1.1)
Breakfast cereals	Model ^1^	1.2 (0.9–1.5)	**1.6 (1.2–2.0)**	1.1 (0.9–1.4)	**1.2 (1.0–1.6)**
	Model ^2^	1.1 (0.9–1.4)	**1.5 (1.2–1.8)**	1.1 (0.9–1.3)	1.2 (1.0–1.5)
Savoury snacks	Model ^1^	0.9 (0.7–1.1)	0.9 (0.7–1.1)	0.8 (0.7–1.0)	**0.8 (0.6–1.0)**
	Model ^2^	1.0 (0.7–1.2)	1.0 (0.8–1.2)	0.9 (0.7–1.1)	0.9 (0.7–1.1)
Confectionery	Model ^1^	1.2 (0.9–1.5)	1.2 (1.0–1.6)	0.8 (0.7–1.0)	0.9 (0.7–1.0)
	Model ^2^	1.2 (0.9–1.5)	1.3 (1.0–1.6)	0.9 (0.7–1.1)	0.9 (0.7–1.1)

^1^ Model adjusted for age; ^2^ Model adjusted for age and socioeconomic status; ^a^ low: <1 h/week (reference group), medium: 1–3 h/week, high: >3 h/week; OR—odds ratio; CI—confidence interval; bold: The odds ratio is significantly different from 1 (reference is low PE with *p* < 0.05).

**Table 4 nutrients-11-01060-t004:** Results of logistic regression analyses between food intake and recommended daily physical activity ^a^.

High intake of Food Group		Boys (*n* = 4557) Daily Physical Activity Level		Girls (*n* = 5073) Daily Physical Activity Level	
		Medium	High	Medium	High
		OR (95% CI)	OR (95% CI)	OR (95% CI)	OR (95% CI)
Soft drinks	Model ^1^	0.8 (0.7–1.0)	1.0 (0.8–1.2)	**0.8 (0.6–1.0)**	0.8 (0.7–1.1)
	Model ^2^	0.9 (0.7–1.1)	1.0 (0.8–1.3)	0.8 (0.7–1.0)	0.9 (0.7–1.1)
Juice	Model ^1^	1.2 (0.9–1.5)	**1.3 (1.0–1.6)**	**1.3 (1.0–1.5)**	1.2 (1.0–1.6)
	Model ^2^	1.2 (0.9–1.5)	**1.3 (1.0–1.6)**	**1.3 (1.1–1.6)**	1.3 (1.0–1.6)
Water	Model ^1^	1.2 (0.9–1.4)	**1.4 (1.1–1.8)**	1.0 (0.9–1.2)	**1.3 (1.0–1.6)**
	Model ^2^	1.2 (1.0–1.5)	**1.4 (1.1–1.8)**	1.1 (0.9–1.3)	**1.3 (1.0–1.6)**
Milk	Model ^1^	**1.3 (1.0–1.7)**	**1.6 (1.2–2.2)**	1.1 (0.9–1.4)	**1.3 (1.0–1.7)**
	Model ^2^	**1.3 (1.0–1.7)**	**1.7 (1.3–2.2)**	1.1 (0.9–1.4)	**1.3 (1.0–1.7)**
Dairy products	Model ^1^	**1.3 (1.1–1.6)**	**1.4 (1.1–1.8)**	**1.2 (1.0–1.5)**	**1.4 (1.1–1.7)**
	Model ^2^	**1.3 (1.1–1.6)**	**1.4 (1.1–1.8)**	1.2 (1.0–1.5)	**1.4 (1.1–1.7)**
Fast food	Model ^1^	1.0 (0.8–1.3)	1.0 (0.8–1.3)	1.0 (0.9–1.3)	0.9 (0.8–1.2)
	Model ^2^	1.0 (0.8–1.3)	1.0 (0.8–1.3)	1.1 (0.9–1.3)	1.0 (0.8–1.2)
Fruits	Model ^1^	**1.9 (1.5–2.3)**	**2.1 (1.7–2.7)**	**1.5 (1.3–1.9)**	**1.9 (1.6–2.4)**
	Model ^2^	**1.8 (1.5–2.3)**	**2.1 (1.6–2.7)**	**1.5 (1.3–1.8)**	**1.9 (1.6–2.3)**
Vegetables	Model ^1^	**1.4 (1.1–1.7)**	**1.6 (1.2–2.0)**	**1.4 (1.2–1.7)**	**1.6 (1.2–2.0)**
	Model ^2^	**1.3 (1.1–1.6)**	**1.5 (1.2–2.0)**	**1.3 (1.1–1.6)**	**1.5 (1.2–1.9)**
Bread	Model ^1^	1.2 (1.0–1.4)	**1.5 (1.2–1.8)**	1.1 (0.9–1.3)	1.2 (1.0–1.6)
	Model ^2^	1.2 (1.0–1.5)	**1.5 (1.2–1.9)**	1.1 (0.9–1.3)	1.3 (1.0–1.6)
Potatoes/pasta/rice	Model ^1^	1.1 (0.8–1.3)	**1.4 (1.1–1.8)**	1.2 (1.0–1.4)	1.1 (0.9–1.3)
	Model ^2^	1.1 (0.8–1.3)	**1.4 (1.1–1.8)**	1.2 (1.0–1.5)	1.1 (0.9–1.3)
Meat	Model ^1^	1.0 (0.8–1.3)	1.3 (1.0–1.6)	1.0 (0.8–1.1)	1.1 (0.9–1.4)
	Model ^2^	1.1 (0.9–1.3)	**1.3 (1.0–1.6)**	1.0 (0.8–1.2)	1.1 (0.9–1.4)
Breakfast cereals	Model ^1^	**1.3 (1.1–1.6)**	**1.4 (1.1–1.7)**	1.1 (0.9–1.3)	1.2 (1.0–1.5)
	Model ^2^	**1.3 (1.0–1.6)**	**1.3 (1.1–1.6)**	1.1 (0.9–1.3)	1.2 (1.0–1.5)
Savoury snacks	Model ^1^	1.0 (0.8–1.2)	1.0 (0.8–1.3)	**0.8 (0.6–0.9)**	0.9 (0.7–1.1)
	Model ^2^	1.1 (0.9–1.3)	1.1 (0.9–1.4)	**0.8 (0.7–1.0)**	0.9 (0.7–1.1)
Confectionery	Model ^1^	1.2 (0.9–1.5)	**1.6 (1.2–2.0)**	**0.8 (0.6–0.9)**	0.9 (0.8-1.1)
	Model ^2^	1.2 (0.9–1.5)	**1.6 (1.2–2.0)**	**0.8 (0.6–0.9)**	1.0 (0.8–1.2)

^1^ Model adjusted for age; ^2^ Model adjusted for age and socioeconomic status; ^a^ low: <3 days/week 1 h/day (reference group), medium: 3–5 days/week 1 h/day, high: 6–7 days/week 1 h/day; OR—odds ratio; CI—confidence interval; bold: The odds ratio is significantly different from 1 (reference is low RDPA with *p* < 0.05).

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
