# Peer review of "Associations between Physical Activity and Food Intake among Children and Adolescents: Results of KiGGS Wave 2"

_nutrients, 2019, doi:10.3390/nu11051060_

Reviewer 1 Report

This paper addresses the relationships between physical activity and food intake among children – it is an important topic, and the paper is well-written and engaging.  I do have a few suggestions/concerns about the paper in its present form:

1.       Given that adults and children have different information about children’s diet and PA (e.g. parents may have partial information on what children eat and do), I would like to see more discussion of the choice to combine parent- and child-reported data within one analysis.  For instance, how may this difference have been confounded with child age in the analysis and interpretation of results?

2.       For TPA, it seems problematic that only the frequency of at least 60 minutes was assessed.  So someone who got 45 minutes every day would be scored as getting less physical activity than someone who got 1hour/day 3 days in a week.  I realize that this is a limitation of the data itself rather than a choice made for this study, but it would be good to mention in the limitations section. Also, it might be more accurate to re-label this variable – it isn’t really total physical activity, so much as frequency of fully meeting daily physical activity recommendations….

3.       For food intake, what is the rationale for coding the raw data into quintiles?  What if the sample overall consumed little of something – a quintile of children who consumed perhaps healthily low levels would nonetheless end up scored as having “high intake”.  Why not use the raw data?  Or, convert into a measure that reflects quality in an absolute way? At least I’d like to see the intake range reported for each quintile.

4.       Table 2 – if the top two quintiles for each food type are assigned as “high intake”, why is only 27.8% of the sample high intake for milk? (Also, a minor point but the upper level headings aren’t centered well, and it makes this table difficult to read).

5.       The sheer number of models that were run is concerning statistically, but moreso, I am concerned that it risks too much detail for the substantive message to come through clearly.  Why not combine fresh fruits and vegetables?  The different forms of grains (cereal, pasta, rice, bread)?  Milk and other dairy products? 

Author Response

Dear Reviewers,

We attach a revised version of our manuscript for review. We thank you for the valuable and thoughtful comments, which improved the quality of our manuscript.  A point by point reply to the comments is attached.The revisions are also highlighted in the manuscript using the “Track Changes” function.

Kind Regards,

the Authors

Reviewer 2 Report

Thanks for submitting this paper for review. I think you’ve done a great job with collating all the data for this. The research is of interest given its novelty and the size of the sample size. I do think the manuscript could be written more concisely in parts, particularly your discussion. I have a made several observations below for you to consider. Line 37. Sedentary behaviour and physical activity are not the opposite of each other. You can be physically active and have a sedentary lifestyle at the same time. Therefore, have a think about that sentence. Line 42: I don’t think you need to give the study's full name here. Line 52-66: On a few occasions towards the end of the intro you speak in present tense, when it's more appropriate to speak in past tense as you're talking about existing work (I.e, studies that have already been carried out). Line 72: Remove the Space between Germany and the full stop. Line 85-87: You've said that the questionnaire was self-administered, so I assume instructions on how to complete it were included? If so, you should mention this. Line 87: You mention in line 93 that the questionnaire includes more items for the older children, so you may what to mention here how the questionnaire was adapted to suit older children. Line 111: I think it makes more sense to use the term "reduced sugar soft drinks" rather than reduced energy soft drinks. Line 124-137: Were the TPA and PE questions taken from a previously validated questionnaire, if so mention this and reference the paper if possible. Also, how was physical activity defined? Also, TPA would technically include light physical activity as well as moderate and vigorous physical activity, but the WHO guidelines are only concerned with MVPA. If you're only interested in MVPA I would change the wording. In addition, please define what you mean by "physical exercise"? Line 190-195: Your tables aren't clearly presented and look a little untidy, particularly table 1. They need some work. Line 196-223. For your multivariable analysis section, you should try to vary how you start your sentences, as at present your wording is a little repetitive. Also, I'm not sure how necessary it is for you to talk about the unadjusted associations in the results, but leave them in if you think they're important to include. Line 222: You don't mention in the methods how you obtained BMI-for age (or at least I don't think you do), was it self reported or objectively measured? Also they're usually referred to as BMI-Z-scores in the literature. Line 241-249. Is it necessary to repeat the results here? Line 225-256. I don’t see what this sentence adds to lines 250-252, and do you mean a “higher energy intake” or a “higher energy requirement”? Line 308-309: Screen-time and sleep duration are lifestyle behaviours, and completely different constructs to physical activity. I would use a better term than “physical activity” to describe the recommendations.   Line 331-334. I think it would better to talk about your strengths first and then move on to your limitations.

Author Response

Dear Reviewer,

We attach a revised version of our manuscript for review. We thank you for the valuable and thoughtful comments, which improved the quality of our manuscript. A point by point reply to the comments is attached. The revisions are also highlighted in the manuscript using the “Track Changes” function.

Kind Regards,

the Authors
